# Robustness of classifiers:
# from adversarial to random noise

**Alhussein Fawzi,**[*] **Seyed-Mohsen Moosavi-Dezfooli,**[*] **Pascal Frossard**
École Polytechnique Fédérale de Lausanne
Lausanne, Switzerland
{alhussein.fawzi, seyed.moosavi, pascal.frossard} at epfl.ch

## Abstract

Several recent works have shown that state-of-the-art classifiers are vulnerable to worst-case (i.e., adversarial) perturbations of the datapoints. On the other hand, it has been empirically observed that these same classifiers are relatively robust to random noise. In this paper, we propose to study a *semi-random* noise regime that generalizes both the random and worst-case noise regimes. We propose the first quantitative analysis of the robustness of nonlinear classifiers in this general noise regime. We establish precise theoretical bounds on the robustness of classifiers in this general regime, which depend on the curvature of the classifier's decision boundary. Our bounds confirm and quantify the empirical observations that classifiers satisfying curvature constraints are robust to random noise. Moreover, we quantify the robustness of classifiers in terms of the subspace dimension in the semi-random noise regime, and show that our bounds remarkably interpolate between the worst-case and random noise regimes. We perform experiments and show that the derived bounds provide very accurate estimates when applied to various state-of-the-art deep neural networks and datasets. This result suggests bounds on the curvature of the classifiers' decision boundaries that we support experimentally, and more generally offers important insights onto the geometry of high dimensional classification problems.

## 1   Introduction

State-of-the-art classifiers, especially deep networks, have shown impressive classification performance on many challenging benchmarks in visual tasks [9] and speech processing [7]. An equally important property of a classifier that is often overlooked is its *robustness* in noisy regimes, when data samples are perturbed by noise. The robustness of a classifier is especially fundamental when it is deployed in real-world, uncontrolled, and possibly hostile environments. In these cases, it is crucial that classifiers exhibit good robustness properties. In other words, a sufficiently small perturbation of a datapoint should ideally not result in altering the estimated label of a classifier. State-of-the-art deep neural networks have recently been shown to be very unstable to worst-case perturbations of the data (or equivalently, *adversarial* perturbations) [17]. In particular, despite the excellent classification performances of these classifiers, well-sought perturbations of the data can easily cause misclassification, since data points often lie very close to the decision boundary of the classifier. Despite the importance of this result, the *worst-case* noise regime that is studied in [17] only represents a very specific type of noise. It furthermore requires the full knowledge of the classification model, which may be a hard assumption in practice.

In this paper, we precisely quantify the robustness of nonlinear classifiers in two practical noise regimes, namely random and semi-random noise regimes. In the *random noise* regime, datapoints are

---

[*]The first two authors contributed equally to this work.

perturbed by noise with random direction in the input space. The *semi-random* regime generalizes this model to random *subspaces* of arbitrary dimension, where a worst-case perturbation is sought within the subspace. In both cases, we derive bounds that precisely describe the robustness of classifiers in function of the *curvature* of the decision boundary. We summarize our contributions as follows:

- In the random regime, we show that the robustness of classifiers behaves as $\sqrt{d}$ times the distance from the datapoint to the classification boundary (where $d$ denotes the dimension of the data) provided the curvature of the decision boundary is sufficiently small. This result highlights the blessing of dimensionality for classification tasks, as it implies that robustness to random noise in high dimensional classification problems can be achieved, even at datapoints that are very close to the decision boundary.

- This quantification notably extends to the general semi-random regime, where we show that the robustness precisely behaves as $\sqrt{d/m}$ times the distance to boundary, with $m$ the dimension of the subspace. This result shows in particular that, even when $m$ is chosen as a small fraction of the dimension $d$, it is still possible to find *small* perturbations that cause data misclassification.

- We empirically show that our theoretical estimates are very accurately satisfied by state-of-the-art deep neural networks on various sets of data. This in turn suggests quantitative insights on the curvature of the decision boundary that we support experimentally through the visualization and estimation on two-dimensional sections of the boundary.

The robustness of classifiers to noise has been the subject of intense research. The robustness properties of SVM classifiers have been studied in [19] for example, and robust optimization approaches for constructing robust classifiers have been proposed to minimize the worst possible empirical error under noise disturbance [1, 10]. More recently, following the recent results on the instability of deep neural networks to worst-case perturbations [17], several works have provided explanations of the phenomenon [3, 5, 14, 18], and designed more robust networks [6, 8, 20, 13, 15, 12]. In [18], the authors provide an interesting empirical analysis of the adversarial instability, and show that adversarial examples are not isolated points, but rather occupy dense regions of the pixel space. In [4], state-of-the-art classifiers are shown to be vulnerable to geometrically constrained adversarial examples. Our work differs from these works, as we provide a theoretical study of the robustness of classifiers to random and semi-random noise in terms of the robustness to adversarial noise. In [3], a formal relation between the robustness to random noise, and the worst-case robustness is established in the case of linear classifiers. Our result therefore generalizes [3] in many aspects, as we study general nonlinear classifiers, and robustness to semi-random noise. Finally, it should be noted that the authors in [5] conjecture that the "high linearity" of classification models explains their instability to adversarial perturbations. The objective and approach we follow here is however different, as we study theoretical relations between the robustness to random, semi-random and adversarial noise.

## 2 Definitions and notations

Let $f : \mathbb{R}^d \to \mathbb{R}^L$ be an $L$-class classifier. Given a datapoint $\boldsymbol{x}_0 \in \mathbb{R}^d$, the estimated label is obtained by $\hat{k}(\boldsymbol{x}_0) = \operatorname{argmax}_k f_k(\boldsymbol{x}_0)$, where $f_k(\boldsymbol{x})$ is the $k^{\text{th}}$ component of $f(\boldsymbol{x})$ that corresponds to the $k^{\text{th}}$ class. Let $\mathcal{S}$ be an arbitrary subspace of $\mathbb{R}^d$ of dimension $m$. Here, we are interested in quantifying the robustness of $f$ with respect to different noise regimes. To do so, we define $\boldsymbol{r}_{\mathcal{S}}^*$ to be the perturbation in $\mathcal{S}$ of minimal norm that is required to change the estimated label of $f$ at $\boldsymbol{x}_0$.[2]

$$\boldsymbol{r}_{\mathcal{S}}^*(\boldsymbol{x}_0) = \operatorname*{argmin}_{\boldsymbol{r} \in \mathcal{S}} \|\boldsymbol{r}\|_2 \text{ s.t. } \hat{k}(\boldsymbol{x}_0 + \boldsymbol{r}) \neq \hat{k}(\boldsymbol{x}_0). \tag{1}$$

Note that $\boldsymbol{r}_{\mathcal{S}}^*(\boldsymbol{x}_0)$ can be equivalently written

$$\boldsymbol{r}_{\mathcal{S}}^*(\boldsymbol{x}_0) = \operatorname*{argmin}_{\boldsymbol{r} \in \mathcal{S}} \|\boldsymbol{r}\|_2 \text{ s.t. } \exists k \neq \hat{k}(\boldsymbol{x}_0) : f_k(\boldsymbol{x}_0 + \boldsymbol{r}) \geq f_{\hat{k}(\boldsymbol{x}_0)}(\boldsymbol{x}_0 + \boldsymbol{r}). \tag{2}$$

When $\mathcal{S} = \mathbb{R}^d$, $\boldsymbol{r}^*(\boldsymbol{x}_0) := \boldsymbol{r}_{\mathbb{R}^d}^*(\boldsymbol{x}_0)$ is the *adversarial (or worst-case) perturbation* defined in [17], which corresponds to the (unconstrained) perturbation of minimal norm that changes the label of the

datapoint $\boldsymbol{x}_0$. In other words, $\|\boldsymbol{r}^*(\boldsymbol{x}_0)\|_2$ corresponds to the minimal distance from $\boldsymbol{x}_0$ to the classifier boundary. In the case where $\mathcal{S} \subset \mathbb{R}^d$, only perturbations along $\mathcal{S}$ are allowed. The robustness of $f$ at $\boldsymbol{x}_0$ along $\mathcal{S}$ is naturally measured by the norm $\|\boldsymbol{r}^*_{\mathcal{S}}(\boldsymbol{x}_0)\|_2$. Different choices for $\mathcal{S}$ permit to study the robustness of $f$ in two different regimes:

- **Random noise regime**: This corresponds to the case where $\mathcal{S}$ is a *one-dimensional subspace* ($m = 1$) with direction $\boldsymbol{v}$, where $\boldsymbol{v}$ is a *random vector* sampled uniformly from the unit sphere $\mathbb{S}^{d-1}$. Writing it explicitly, we study in this regime the robustness quantity defined by $\min_t |t|$ s.t. $\exists k \neq \hat{k}(\boldsymbol{x}_0), f_k(\boldsymbol{x}_0 + t\boldsymbol{v}) \geq f_{\hat{k}(\boldsymbol{x}_0)}(\boldsymbol{x}_0 + t\boldsymbol{v})$, where $\boldsymbol{v}$ is a vector sampled uniformly at random from the unit sphere $\mathbb{S}^{d-1}$.
- **Semi-random noise regime**: In this case, the subspace $\mathcal{S}$ is chosen *randomly*, but can be of arbitrary dimension $m$.[3] We use the *semi*-random terminology as the subspace is chosen randomly, and the smallest vector that causes misclassification is then sought in the subspace. It should be noted that the random noise regime is a special case of the semi-random regime with a subspace of dimension $m = 1$. We differentiate nevertheless between these two regimes for clarity.

In the remainder of the paper, the goal is to establish relations between the robustness in the random and semi-random regimes on the one hand, and the robustness to adversarial perturbations $\|\boldsymbol{r}^*(\boldsymbol{x}_0)\|_2$ on the other hand. We recall that the latter quantity captures the distance from $\boldsymbol{x}_0$ to the classifier boundary, and is therefore a key quantity in the analysis of robustness.

In the following analysis, we fix $\boldsymbol{x}_0$ to be a datapoint classified as $\hat{k}(\boldsymbol{x}_0)$. To simplify the notation, we remove the explicit dependence on $\boldsymbol{x}_0$ in our notations (e.g., we use $\boldsymbol{r}^*_{\mathcal{S}}$ instead of $\boldsymbol{r}^*_{\mathcal{S}}(\boldsymbol{x}_0)$ and $\hat{k}$ instead of $\hat{k}(\boldsymbol{x}_0)$), and it should be implicitly understood that all our quantities pertain to the fixed datapoint $\boldsymbol{x}_0$.

## 3  Robustness of affine classifiers

We first assume that $f$ is an affine classifier, i.e., $f(\boldsymbol{x}) = \mathbf{W}^\top \boldsymbol{x} + \boldsymbol{b}$ for a given $\mathbf{W} = [\boldsymbol{w}_1 \dots \boldsymbol{w}_L]$ and $\boldsymbol{b} \in \mathbb{R}^L$.

The following result shows a precise relation between the robustness to semi-random noise, $\|\boldsymbol{r}^*_{\mathcal{S}}\|_2$ and the robustness to adversarial perturbations, $\|\boldsymbol{r}^*\|_2$.

**Theorem 1.** *Let $\delta > 0$, $\mathcal{S}$ be a random $m$-dimensional subspace of $\mathbb{R}^d$, and $f$ be a $L$-class affine classifier. Let*

$$\zeta_1(m, \delta) = \left( 1 + 2\sqrt{\frac{\ln(1/\delta)}{m}} + \frac{2\ln(1/\delta)}{m} \right)^{-1}, \tag{3}$$

$$\zeta_2(m, \delta) = \left( \max \left( (1/e)\delta^{2/m}, 1 - \sqrt{2(1 - \delta^{2/m})} \right) \right)^{-1}. \tag{4}$$

*The following inequalities hold between the robustness to semi-random noise $\|\boldsymbol{r}^*_{\mathcal{S}}\|_2$, and the robustness to adversarial perturbations $\|\boldsymbol{r}^*\|_2$:*

$$\sqrt{\zeta_1(m, \delta)}\sqrt{\frac{d}{m}}\|\boldsymbol{r}^*\|_2 \leq \|\boldsymbol{r}^*_{\mathcal{S}}\|_2 \leq \sqrt{\zeta_2(m, \delta)}\sqrt{\frac{d}{m}}\|\boldsymbol{r}^*\|_2, \tag{5}$$

*with probability exceeding $1 - 2(L + 1)\delta$.*

The proof can be found in the appendix. Our upper and lower bounds depend on the functions $\zeta_1(m, \delta)$ and $\zeta_2(m, \delta)$ that control the inequality constants (for $m$, $\delta$ fixed). It should be noted that $\zeta_1(m, \delta)$ and $\zeta_2(m, \delta)$ are independent of the data dimension $d$. Fig. 1 shows the plots of $\zeta_1(m, \delta)$ and $\zeta_2(m, \delta)$ as functions of $m$, for a fixed $\delta$. It should be noted that for sufficiently large $m$, $\zeta_1(m, \delta)$ and $\zeta_2(m, \delta)$ are very close to 1 (e.g., $\zeta_1(m, \delta)$ and $\zeta_2(m, \delta)$ belong to the interval $[0.8, 1.3]$ for $m \geq 250$ in the settings of Fig. 1). The interval $[\zeta_1(m, \delta), \zeta_2(m, \delta)]$ is however (unavoidably) larger when $m = 1$.

The result in Theorem 1 shows that in the random and semi-random noise regimes, the robustness to noise is precisely related to $\|\boldsymbol{r}^*\|_2$ by a factor of $\sqrt{d/m}$. Specifically, in the random noise regime ($m = 1$), the magnitude of the noise required to misclassify the datapoint behaves as $\Theta(\sqrt{d}\|\boldsymbol{r}^*\|_2)$ with high probability, with constants in the interval $[\zeta_1(1,\delta), \zeta_2(1,\delta)]$. Our results therefore show that, in high dimensional classification settings, affine classifiers can be robust to random noise, even if the datapoint lies very closely to the decision boundary (i.e., $\|\boldsymbol{r}^*\|_2$ is small). In the semi-random noise regime with $m$ sufficiently large (e.g., $m \geq 250$), we have $\|\boldsymbol{r}_{\mathcal{S}}^*\|_2 \approx \sqrt{d/m}\|\boldsymbol{r}^*\|_2$ with high probability, as the constants $\zeta_1(m,\delta) \approx \zeta_2(m,\delta) \approx 1$ for sufficiently large $m$. Our bounds therefore "interpolate" between the random

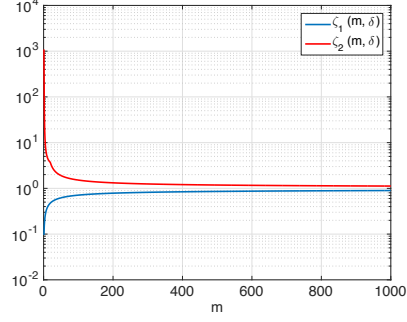

Figure 1: $\zeta_1(m,\delta)$ and $\zeta_2(m,\delta)$ in function of $m$ [$\delta = 0.05$] .

noise regime, which behaves as $\sqrt{d}\|\boldsymbol{r}^*\|_2$, and the worst-case noise $\|\boldsymbol{r}^*\|_2$. More importantly, the square root dependence is also notable here, as it shows that the semi-random robustness can remain small even in regimes where $m$ is chosen to be a very small fraction of $d$. For example, choosing a small subspace of dimension $m = 0.01d$ results in semi-random robustness of $10\|\boldsymbol{r}^*\|_2$ with high probability, which might still not be perceptible in complex visual tasks. Hence, for semi-random noise that is mostly random and only mildly adversarial (i.e., the subspace dimension is small), affine classifiers remain vulnerable to such noise.

## 4 Robustness of general classifiers

### 4.1 Curvature of the decision boundary

We now consider the general case where $f$ is a nonlinear classifier. We derive relations between the random and semi-random robustness $\|\boldsymbol{r}_{\mathcal{S}}^*\|_2$ and worst-case robustness $\|\boldsymbol{r}^*\|_2$ using properties of the classifier's *boundary*. Let $i$ and $j$ be two arbitrary classes; we define the pairwise boundary $\mathscr{B}_{i,j}$ as the boundary of the *binary* classifier where only classes $i$ and $j$ are considered. Formally, the decision boundary is given by $\mathscr{B}_{i,j} := \{\boldsymbol{x} \in \mathbb{R}^d : f_i(\boldsymbol{x}) - f_j(\boldsymbol{x}) = 0\}$. The boundary $\mathscr{B}_{i,j}$ separates between two regions of $\mathbb{R}^d$, namely $\mathcal{R}_i$ and $\mathcal{R}_j$, where the estimated label of the binary classifier is respectively $i$ and $j$.

We assume for the purpose of this analysis that the boundary $\mathscr{B}_{i,j}$ is smooth. We are now interested in the geometric properties of the boundary, namely its curvature. Many notions of curvature can be defined on hypersurfaces [11]. In the simple case of a curve in a two-dimensional space, the curvature is defined as the inverse of the radius of the so-called oscullating circle. One way to define curvature for high-dimensional hypersurfaces is by taking *normal* sections of the hypersurface, and measuring the curvature of the resulting planar curve (see Fig. 2). We however introduce a notion of curvature that is specifically suited to the analysis of the decision boundary of a classifier. Informally, our curvature captures the *global* bending of the decision boundary by inscribing balls in the regions separated by the decision boundary. For a given $\boldsymbol{p} \in \mathscr{B}_{i,j}$, we define $q_{i \parallel j}(\boldsymbol{p})$ to be the radius of the largest open ball included in the region $\mathcal{R}_i$ that intersects with $\mathscr{B}_{i,j}$ at $\boldsymbol{p}$; i.e.,

$$q_{i \parallel j}(\boldsymbol{p}) = \sup_{\boldsymbol{z} \in \mathbb{R}^d} \{\|\boldsymbol{z} - \boldsymbol{p}\|_2 : B(\boldsymbol{z}, \|\boldsymbol{z} - \boldsymbol{p}\|_2) \subseteq \mathcal{R}_i\}, \qquad (6)$$

where $B(\boldsymbol{z}, \|\boldsymbol{z} - \boldsymbol{p}\|_2)$ is the open ball in $\mathbb{R}^d$ of center $\boldsymbol{z}$ and radius $\|\boldsymbol{z} - \boldsymbol{p}\|_2$. An illustration of this quantity in two dimensions is provided in Fig. 2 (b). It is not hard to see that any ball $B(\boldsymbol{z}^*, \|\boldsymbol{z}^* - \boldsymbol{p}\|_2)$ centered in $\boldsymbol{z}^*$ and included in $\mathcal{R}_i$ will have its tangent space at $\boldsymbol{p}$ coincide with the tangent of the decision boundary at the same point.

It should further be noted that the definition in Eq. (6) is not symmetric in $i$ and $j$. We therefore define the following symmetric quantity $q_{i,j}(\boldsymbol{p})$, where the worst-case ball inscribed in any of the two regions $\mathcal{R}_i$ and $\mathcal{R}_j$ is considered:

$$q_{i,j}(\boldsymbol{p}) = \min\left(q_{i \parallel j}(\boldsymbol{p}), q_{j \parallel i}(\boldsymbol{p})\right).$$

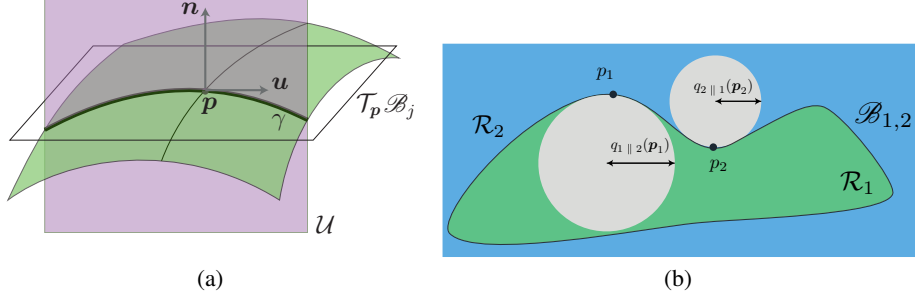

(a)    (b)

Figure 2: (a) Normal section of the boundary $\mathscr{B}_{i,j}$ with respect to plane $\mathcal{U} = \text{span}(\boldsymbol{n}, \boldsymbol{u})$, where $\boldsymbol{n}$ is the normal to the boundary at $\boldsymbol{p}$, and $\boldsymbol{u}$ is an arbitrary in the tangent space $\mathcal{T}_{\boldsymbol{p}}(\mathscr{B}_{i,j})$. (b) Illustration of the quantities introduced for the definition of the curvature of the decision boundary.

To measure the global curvature, the worst-case radius is taken over all points on the decision boundary, i.e., $q(\mathscr{B}_{i,j}) = \inf_{\boldsymbol{p} \in \mathscr{B}_{i,j}} q_{i,j}(\boldsymbol{p})$. The curvature $\kappa(\mathscr{B}_{i,j})$ is then defined as the inverse of the worst-case radius: $\kappa(\mathscr{B}_{i,j}) = 1/q(\mathscr{B}_{i,j})$.

In the case of affine classifiers, we have $\kappa(\mathscr{B}_{i,j}) = 0$, as it is possible to inscribe balls of infinite radius inside each region of the space. When the classification boundary is a union of (sufficiently distant) spheres with equal radius $R$, the curvature $\kappa(\mathscr{B}_{i,j}) = 1/R$. In general, the quantity $\kappa(\mathscr{B}_{i,j})$ provides an intuitive way of describing the nonlinearity of the decision boundary by fitting balls inside the classification regions.

## 4.2    Robustness to random and semi-random noise

We now establish bounds on the robustness to random and semi-random noise in the binary classification case. Let $\boldsymbol{x}_0$ be a datapoint classified as $\hat{k} = \hat{k}(\boldsymbol{x}_0)$. We first study the binary classification problem, where only classes $\hat{k}$ and $k \in \{1, \dots, L\} \setminus \{\hat{k}\}$ are considered. To simplify the notation, we let $\mathscr{B}_k := \mathscr{B}_{k,\hat{k}}$ be the decision boundary between classes $k$ and $\hat{k}$. In the case of the binary classification problem where classes $k$ and $\hat{k}$ are considered, the semi-random perturbation defined in Eq. (2) can be re-written as follows:

$$\boldsymbol{r}_{\mathcal{S}}^k = \underset{\boldsymbol{r} \in \mathcal{S}}{\arg\min} \|\boldsymbol{r}\|_2 \text{ s.t. } f_k(\boldsymbol{x}_0 + \boldsymbol{r}) \geq f_{\hat{k}}(\boldsymbol{x}_0 + \boldsymbol{r}). \tag{7}$$

The worst case perturbation (obtained with $\mathcal{S} = \mathbb{R}^d$) is denoted by $\boldsymbol{r}^k$. It should be noted that the global quantities $\boldsymbol{r}_{\mathcal{S}}^*$ and $\boldsymbol{r}^*$ are obtained from $\boldsymbol{r}_{\mathcal{S}}^k$ and $\boldsymbol{r}^k$ by taking the vectors with minimum norm over all classes $k$.

The following result gives upper and lower bounds on the ratio $\frac{\|\boldsymbol{r}_{\mathcal{S}}^k\|_2}{\|\boldsymbol{r}^k\|_2}$ in function of the curvature of the boundary separating class $k$ and $\hat{k}$.

**Theorem 2.** *Let $\mathcal{S}$ be a random $m$-dimensional subspace of $\mathbb{R}^d$. Let $\kappa := \kappa(\mathscr{B}_k)$. Assuming that the curvature satisfies*

$$\kappa \leq \frac{C}{\zeta_2(m,\delta)\|\boldsymbol{r}^k\|_2} \frac{m}{d}, \tag{8}$$

*the following inequality holds between the semi-random robustness $\|\boldsymbol{r}_{\mathcal{S}}^k\|_2$ and the adversarial robustness $\|\boldsymbol{r}^k\|_2$:*

$$\left(1 - C_1\|\boldsymbol{r}^k\|_2 \kappa \zeta_2 \frac{d}{m}\right) \sqrt{\zeta_1} \sqrt{\frac{d}{m}} \leq \frac{\|\boldsymbol{r}_{\mathcal{S}}^k\|_2}{\|\boldsymbol{r}^k\|_2} \leq \left(1 + C_2\|\boldsymbol{r}^k\|_2 \kappa \zeta_2 \frac{d}{m}\right) \sqrt{\zeta_2} \sqrt{\frac{d}{m}} \tag{9}$$

*with probability larger than $1 - 4\delta$. We recall that $\zeta_1 = \zeta_1(m,\delta)$ and $\zeta_2 = \zeta_2(m,\delta)$ are defined in Eq. (3, 4). The constants are $C = 0.2, C_1 = 0.625, C_2 = 2.25$.*

The proof can be found in the appendix. This result shows that the bounds relating the robustness to random and semi-random noise to the worst-case robustness can be extended to nonlinear classifiers,

provided the curvature of the boundary $\kappa(\mathscr{B}_k)$ is sufficiently small. In the case of linear classifiers, we have $\kappa(\mathscr{B}_k) = 0$, and we recover the result for affine classifiers from Theorem 1.

To extend this result to multi-class classification, special care has to be taken. In particular, if $k$ denotes a class that has no boundary with class $\hat{k}$, $\|\boldsymbol{r}^k\|_2$ can be very large and the previous curvature condition is not satisfied. It is therefore crucial to *exclude* such classes that have no boundary in common with class $\hat{k}$, or more generally, boundaries that are far from class $\hat{k}$. We define the set $A$ of excluded classes $k$ where $\|\boldsymbol{r}^k\|_2$ is large

$$A = \{k : \|\boldsymbol{r}^k\|_2 \geq 1.45\sqrt{\zeta_2(m,\delta)}\sqrt{\frac{d}{m}}\|\boldsymbol{r}^*\|_2\}. \tag{10}$$

Note that $A$ is independent of $\mathcal{S}$, and depends only on $d$, $m$ and $\delta$. Moreover, the constants in (10) were chosen for simplicity of exposition.

Assuming a curvature constraint *only on the close enough classes*, the following result establishes a simplified relation between $\|\boldsymbol{r}_\mathcal{S}^*\|_2$ and $\|\boldsymbol{r}^*\|_2$.

**Corollary 1.** *Let $\mathcal{S}$ be a random $m$-dimensional subspace of $\mathbb{R}^d$. Assume that, for all $k \notin A$, the curvature condition in Eq. (8) holds. Then, we have*

$$0.875\sqrt{\zeta_1(m,\delta)}\sqrt{\frac{d}{m}}\|\boldsymbol{r}^*\|_2 \leq \|\boldsymbol{r}_\mathcal{S}^*\|_2 \leq 1.45\sqrt{\zeta_2(m,\delta)}\sqrt{\frac{d}{m}}\|\boldsymbol{r}^*\|_2 \tag{11}$$

*with probability larger than $1 - 4(L+2)\delta$.*

Under the curvature condition in (8) on the boundaries between $\hat{k}$ and classes in $A^c$, our result shows that the robustness to random and semi-random noise exhibits the same behavior that has been observed earlier for linear classifiers in Theorem 1. In particular, $\|\boldsymbol{r}_\mathcal{S}^*\|_2$ is precisely related to the adversarial robustness $\|\boldsymbol{r}^*\|_2$ by a factor of $\sqrt{d/m}$. In the random regime ($m = 1$), this factor becomes $\sqrt{d}$, and shows that in high dimensional classification problems, classifiers with sufficiently flat boundaries are much more robust to random noise than to adversarial noise. However, in the semi-random, the factor is $\sqrt{d/m}$ and shows that robustness to semi-random noise might not be achieved even if $m$ is chosen to be a tiny fraction of $d$. In other words, if a classifier is highly vulnerable to adversarial perturbations, then it is also vulnerable to noise that is overwhelmingly random and only mildly adversarial.

It is important to note that the curvature condition in Corollary 1 is *not* an assumption on the curvature of the global decision boundary, but rather an assumption on the decision boundaries between pairs of classes. The distinction here is significant, as junction points where two decision boundaries meet might actually have a very large (or infinite) curvature (even in linear classification settings), and the curvature condition in Corollary 1 typically does not hold for this global curvature definition. We refer to our experimental section for a visualization of this phenomenon.

## 5 Experiments

We now evaluate the robustness of different image classifiers to random and semi-random perturbations, and assess the accuracy of our bounds on various datasets and state-of-the-art classifiers. Specifically, our theoretical results show that the robustness $\|\boldsymbol{r}_\mathcal{S}^*(\boldsymbol{x})\|_2$ of classifiers satisfying the curvature property precisely behaves as $\sqrt{d/m}\|\boldsymbol{r}^*(\boldsymbol{x})\|_2$. We first check the accuracy of these results in different classification settings. For a given classifier $f$ and subspace dimension $m$, we define $\beta(f;m) = \sqrt{m/d}\frac{1}{|\mathscr{D}|}\sum_{\boldsymbol{x}\in\mathscr{D}}\frac{\|\boldsymbol{r}_\mathcal{S}^*(\boldsymbol{x})\|_2}{\|\boldsymbol{r}^*(\boldsymbol{x})\|_2}$, where $\mathcal{S}$ is chosen randomly for each sample $\boldsymbol{x}$ and $\mathscr{D}$ denotes the test set. This quantity provides indication to the accuracy of our $\sqrt{d/m}\|\boldsymbol{r}^*(\boldsymbol{x})\|_2$ estimate of the robustness, and should ideally be equal to 1 (for sufficiently large $m$). Since $\beta$ is a random quantity (because of $\mathcal{S}$), we report both its mean and standard deviation for different networks in Table 1. It should be noted that finding $\|\boldsymbol{r}_\mathcal{S}^*\|_2$ and $\|\boldsymbol{r}^*\|_2$ involves solving the optimization problem in (1). We have used a similar approach to [13] to find subspace minimal perturbations. For each network, we estimate the expectation by averaging $\beta(f;m)$ on 1000 random samples, with $\mathcal{S}$ also chosen randomly for each sample. Observe that $\beta$ is suprisingly close to 1, even when $m$ is a small fraction of $d$. This shows that our quantitative analysis provide very accurate estimates of the robustness to semi-random noise. We visualize the robustness to random noise, semi-random noise (with $m = 10$)

Table 1: $\beta(f; m)$ for different classifiers $f$ and different subspace dimensions $m$. The VGG-F and VGG-19 are respectively introduced in [2, 16].

| Classifier | $m/d$ | | | | |
| --- | --- | --- | --- | --- | --- |
| | 1/4 | 1/16 | 1/36 | 1/64 | 1/100 |
| LeNet (MNIST) | $1.00 \pm 0.06$ | $1.01 \pm 0.12$ | $1.03 \pm 0.20$ | $1.01 \pm 0.26$ | $1.05 \pm 0.34$ |
| LeNet (CIFAR-10) | $1.01 \pm 0.03$ | $1.02 \pm 0.07$ | $1.04 \pm 0.10$ | $1.06 \pm 0.14$ | $1.10 \pm 0.19$ |
| VGG-F (ImageNet) | $1.00 \pm 0.01$ | $1.02 \pm 0.02$ | $1.03 \pm 0.04$ | $1.03 \pm 0.05$ | $1.04 \pm 0.06$ |
| VGG-19 (ImageNet) | $1.00 \pm 0.01$ | $1.02 \pm 0.03$ | $1.02 \pm 0.05$ | $1.03 \pm 0.06$ | $1.04 \pm 0.08$ |

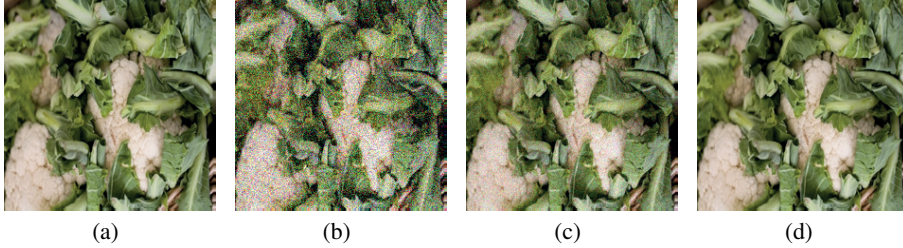

|   (a)   |   (b)   |   (c)   |   (d)   |

Figure 3: (a) Original image classified as "Cauliflower". Fooling perturbations for VGG-F network: (b) Random noise, (c) Semi-random perturbation with $m = 10$, (d) Worst-case perturbation, all wrongly classified as "Artichoke".

and worst-case perturbations on a sample image in Fig. 3. While random noise is clearly perceptible due to the $\sqrt{d} \approx 400$ factor, semi-random noise becomes much less perceptible even with a relatively small value of $m = 10$, thanks to the $1/\sqrt{m}$ factor that attenuates the required noise to misclassify the datapoint. It should be noted that the robustness of neural networks to adversarial perturbations has previously been observed empirically in [17], but we provide here a quantitative and generic explanation for this phenomenon. The high accuracy of our bounds for different state-of-the-art classifiers, and different datasets suggest that the decision boundaries of these classifiers have limited curvature $\kappa(\mathscr{B}_k)$, as this is a key assumption of our theoretical findings. To support the validity of this curvature hypothesis in practice, we visualize two-dimensional sections of the classifiers' boundary in Fig. 4 in three different settings. Note that we have opted here for a visualization strategy rather than the numerical estimation of $\kappa(\mathscr{B})$, as the latter quantity is difficult to approximate in practice in high dimensional problems. In Fig. 4, $\boldsymbol{x}_0$ is chosen randomly from the test set for each data set, and the decision boundaries are shown in the plane spanned by $\boldsymbol{r}^*$ and $\boldsymbol{r}_{\mathcal{S}}^*$, where $\mathcal{S}$ is a random *direction* (i.e., $m = 1$). Different colors on the boundary correspond to boundaries with different classes. It can be observed that the curvature of the boundary is very small except at "junction" points where the boundary of two different classes intersect. Our curvature assumption, which only assumes a bound on the curvature of the decision boundary between pairs of classes $\hat{k}(\boldsymbol{x}_0)$ and $k$ (but not on the *global* decision boundary that contains junctions with high curvature) is therefore adequate to the decision boundaries of state-of-the-art classifiers according to Fig. 4. Interestingly, the assumption in Corollary 1 is satisfied by taking $\kappa$ to be an empirical estimate of the curvature of the planar curves in Fig. 4 (a) for the dimension of the subspace being a *very* small fraction of $d$; e.g., $m = 10^{-3}d$. While not reflecting the curvature $\kappa(\mathscr{B}_k)$ that drives the assumption of our theoretical analysis, this result still seems to suggest that the curvature assumption holds in practice.

We now show a simple demonstration of the vulnerability of classifiers to semi-random noise in Fig. 5, where a structured message is hidden in the image and causes data misclassification. Specifically, we consider $\mathcal{S}$ to be the span of random translated and scaled versions of words "NIPS", "SPAIN" and "2016" in an image, such that $\lfloor d/m \rfloor = 228$. The resulting perturbations in the subspace are therefore linear combinations of these words with different intensities.[4] The perturbed image $\boldsymbol{x}_0 + \boldsymbol{r}_{\mathcal{S}}^*$ shown in

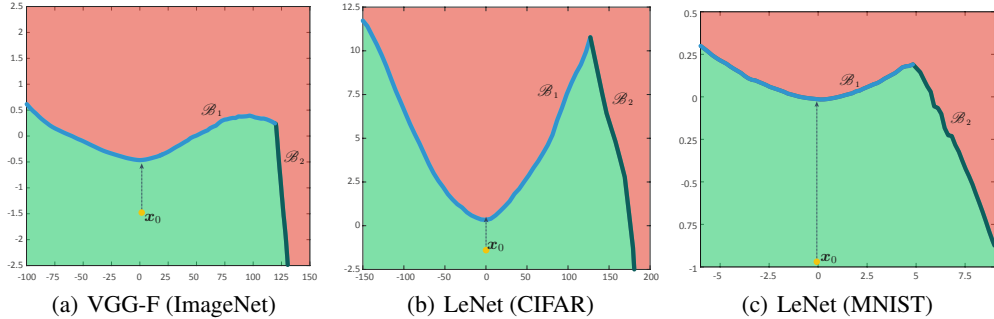

(a) VGG-F (ImageNet)    (b) LeNet (CIFAR)    (c) LeNet (MNIST)

Figure 4: Boundaries of three classifiers near randomly chosen samples. Axes are normalized by the corresponding $\|\boldsymbol{r}^*\|_2$ as our assumption in the theoretical bound depends on the product of $\|\boldsymbol{r}^*\|_2 \kappa$. Note the difference in range between $x$ and $y$ axes. Note also that the range of horizontal axis in (c) is much smaller than the other two, hence the illustrated boundary is more curved.

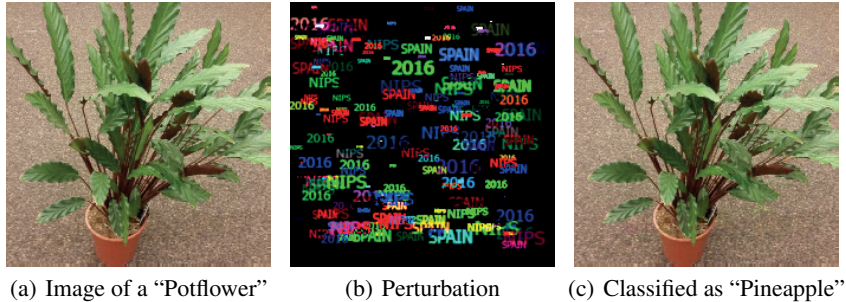

(a) Image of a "Potflower"    (b) Perturbation    (c) Classified as "Pineapple"

Figure 5: A fooling hidden message. $\mathcal{S}$ is the span of random translations and scales of the words "NIPS", "SPAIN", and "2016".

Fig. 5 (c) is clearly indistinguishable from Fig. 5 (a). This shows that imperceptibly small structured messages can be added to an image causing data misclassification.

# 6  Conclusion

In this work, we precisely characterized the robustness of classifiers in a novel semi-random noise regime that generalizes the random noise regime. Specifically, our bounds relate the robustness in this regime to the robustness to adversarial perturbations. Our bounds depend on the *curvature* of the decision boundary, the data dimension, and the dimension of the subspace to which the perturbation belongs. Our results show, in particular, that when the decision boundary has a small curvature, classifiers are robust to random noise in high dimensional classification problems (even if the robustness to adversarial perturbations is relatively small). Moreover, for semi-random noise that is mostly random and only mildly adversarial (i.e., the subspace dimension is small), our results show that state-of-the-art classifiers remain vulnerable to such perturbations. To improve the robustness to semi-random noise, our analysis encourages to impose geometric constraints on the curvature of the decision boundary, as we have shown the existence of an intimate relation between the robustness of classifiers and the curvature of the decision boundary.

**Acknowledgments**

We would like to thank the anonymous reviewers for their helpful comments. We thank Omar Fawzi and Louis Merlin for the fruitful discussions. We also gratefully acknowledge the support of NVIDIA Corporation with the donation of the Tesla K40 GPU used for this research. This work has been partly supported by the Hasler Foundation, Switzerland, in the framework of the CORA project.

## Footnotes

[2]Perturbation vectors sending a datapoint exactly to the boundary are assumed to change the estimated label of the classifier.

[3] A random subspace is defined as the span of $m$ independent vectors drawn uniformly at random from $\mathbb{S}^{d-1}$.

[4]This example departs somehow from the theoretical framework of this paper, where *random* subspaces were considered. However, this empirical example suggests that the theoretical findings in this paper seem to approximately hold when the subspace $\mathcal{S}$ have statistics that are close to a random subspace.

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
