[Supplementary Material]

# Robustness of classifiers: from adversarial to random noise (Supplementary material)

**Alhussein Fawzi**[*]**, Seyed-Mohsen Moosavi-Dezfooli**[*]**, Pascal Frossard**
École Polytechnique Fédérale de Lausanne
Lausanne, Switzerland
{alhussein.fawzi, seyed.moosavi, pascal.frossard} at epfl.ch

## A.1 Proof of Theorem 1 (affine classifiers)

**Lemma 1** ([1])**.** *Let $Y$ be a point chosen uniformly at random from the surface of the d-dimensional sphere $\mathbb{S}^{d-1}$. Let the vector $Z$ be the projection of $Y$ onto its first $m$ coordinates, with $m < d$. Then,*

   *1. If $\beta < 1$, then*

$$\mathbb{P}\left(\|Z\|_2^2 \leq \frac{\beta m}{d}\right) \leq \beta^{m/2}\left(1 + \frac{(1-\beta)m}{(d-m)}\right)^{(d-m)/2} \leq \exp\left(\frac{m}{2}(1-\beta+\ln\beta)\right).$$
(A.1)

   *2. If $\beta > 1$, then*

$$\mathbb{P}\left(\|Z\|_2^2 \geq \frac{\beta m}{d}\right) \leq \beta^{m/2}\left(1 + \frac{(1-\beta)m}{(d-m)}\right)^{(d-m)/2} \leq \exp\left(\frac{m}{2}(1-\beta+\ln\beta)\right).$$
(A.2)

**Lemma 2.** *Let $\boldsymbol{v}$ be a random vector uniformly drawn from the unit sphere $\mathbb{S}^{d-1}$, and $\mathbf{P}_m$ be the projection matrix onto the first $m$ coordinates. Then,*

$$\mathbb{P}\left(\beta_1(\delta,m)\frac{m}{d} \leq \|\mathbf{P}_m\boldsymbol{v}\|_2^2 \leq \beta_2(\delta,m)\frac{m}{d}\right) \geq 1 - 2\delta,$$
(A.3)

*with $\beta_1(\delta,m) = \max((1/e)\delta^{2/m}, 1 - \sqrt{2(1-\delta^{2/m})})$, and $\beta_2(\delta,m) = 1 + 2\sqrt{\frac{\ln(1/\delta)}{m}} + \frac{2\ln(1/\delta)}{m}$.*

*Proof.* Note first that the upper bound of Lemma 1 can be bounded as follows:

$$\beta^{m/2}\left(1 + \frac{(1-\beta)m}{d-m}\right)^{(d-m)/2} \leq \beta^{m/2}\exp\left(\frac{(1-\beta)m}{2}\right),$$
(A.4)

using $1 + x \leq \exp(x)$. We find $\beta$ such that $\beta^{m/2}\exp\left(\frac{(1-\beta)m}{2}\right) \leq \delta$, or equivalently, $\beta\exp(1-\beta) \leq \delta^{2/m}$. It is easy to see that when $\beta = \frac{1}{e}\delta^{2/m}$, the inequality holds. Note however that $\frac{1}{e}\delta^{2/m}$ does not converge to 1 as $m \to \infty$. We therefore need to derive a tighter bound for this regime. Using the inequality $\beta\exp(1-\beta) \leq 1 - \frac{1}{2}(1-\beta)^2$ for $0 \leq \beta \leq 1$, it follows that the inequality $\beta\exp(1-\beta) \leq \delta^{2/m}$ holds for $\beta = 1 - \sqrt{2(1-\delta^{2/m})}$. In this case, we have $1 - \sqrt{2(1-\delta^{2/m})} \to 1$, as $m \to \infty$. We take our lower bound to be the max of both derived bounds (the latter is more appropriate for large $m$, whereas the former is tighter for small $m$).

---

[*]The first two authors contributed equally to this work.

For $\beta_2$, note that the requirement $\beta \exp(1 - \beta) \leq \delta^{2/m}$ is equivalent to $-\ln(\beta) + (\beta - 1) \geq \frac{2}{m} \ln(1/\delta)$. By setting $\beta = \beta_2(\delta, m)$, this condition is equivalent to $2\sqrt{\frac{\ln(1/\delta)}{m}} - \ln(\beta_2(\delta, m)) \geq 0$, or equivalently, $2z - \ln(1 + 2z + 2z^2) \geq 0$, with $z = \sqrt{\frac{\ln(1/\delta)}{m}}$. The function $z \mapsto 2z - \ln(1 + 2z + 2z^2) \geq 0$ is positive on $\mathbb{R}^+$. Hence, $\beta_2(\delta, m)$ satisfies $\beta \exp(1 - \beta) \leq \delta^{2/m}$, which concludes the proof. $\quad\square$

We now prove our main theorem that we recall as follows:

**Theorem 1.** *Let $\mathcal{S}$ be a random $m$-dimensional subspace of $\mathbb{R}^d$. The following inequalities hold between the norms of semi-random perturbation $\boldsymbol{r}_{\mathcal{S}}^*$ and the worst-case perturbation $\boldsymbol{r}^*$. Let $\zeta_1(m, \delta) = \frac{1}{\beta_2(m,\delta)}$, and $\zeta_2(m, \delta) = \frac{1}{\beta_1(m,\delta)}$.*

$$\zeta_1(m, \delta) \frac{d}{m} \|\boldsymbol{r}^*\|_2^2 \leq \|\boldsymbol{r}_{\mathcal{S}}^*\|_2^2 \leq \zeta_2(m, \delta) \frac{d}{m} \|\boldsymbol{r}^*\|_2^2, \tag{A.5}$$

*with probability exceeding $1 - 2(L + 1)\delta$.*

*Proof.* For the linear case, $\boldsymbol{r}^*$ and $\boldsymbol{r}_{\mathcal{S}}^*$ can be computed in closed form. We recall that, for any subspace $\mathcal{S}$, we have

$$\boldsymbol{r}_{\mathcal{S}}^k = \frac{\left| f_k(\boldsymbol{x}_0) - f_{\hat{k}(\boldsymbol{x}_0)}(\boldsymbol{x}_0) \right|}{\|\mathbf{P}_{\mathcal{S}}\boldsymbol{w}_k - \mathbf{P}_{\mathcal{S}}\boldsymbol{w}_{\hat{k}(\boldsymbol{x}_0)}\|_2^2} (\mathbf{P}_{\mathcal{S}}\boldsymbol{w}_k - \mathbf{P}_{\mathcal{S}}\boldsymbol{w}_{\hat{k}(\boldsymbol{x}_0)}), \tag{A.6}$$

where $\boldsymbol{r}_{\mathcal{S}}^k$ was defined in Eq. (7) in the main paper. In particular, when $\mathcal{S} = \mathbb{R}^d$, we have

$$\boldsymbol{r}^k = \frac{\left| f_k(\boldsymbol{x}_0) - f_{\hat{k}(\boldsymbol{x}_0)}(\boldsymbol{x}_0) \right|}{\|\boldsymbol{w}_k - \boldsymbol{w}_{\hat{k}(\boldsymbol{x}_0)}\|_2^2} (\boldsymbol{w}_k - \boldsymbol{w}_{\hat{k}(\boldsymbol{x}_0)}). \tag{A.7}$$

Let $k \neq \hat{k}(\boldsymbol{x}_0)$. Define, for the sake of readability

$$f^k = \left| f_k(\boldsymbol{x}_0) - f_{\hat{k}(\boldsymbol{x}_0)}(\boldsymbol{x}_0) \right|,$$

$$\boldsymbol{z}^k = \boldsymbol{w}_k - \boldsymbol{w}_{\hat{k}(\boldsymbol{x}_0)}.$$

Note that

$$\frac{\|\boldsymbol{r}^k\|_2^2}{\|\boldsymbol{r}_{\mathcal{S}}^k\|_2^2} = \frac{\|\mathbf{P}_{\mathcal{S}}\boldsymbol{z}^k\|_2^2}{\|\boldsymbol{z}^k\|_2^2}. \tag{A.8}$$

The projection of a fixed vector in $\mathbb{S}^{d-1}$ onto a random $m$ dimensional subspace is equivalent (up to a unitary transformation $\mathbf{U}$) to the projection of a random vector uniformly sampled from $\mathbb{S}^{d-1}$ into a fixed subspace. Let $\mathbf{P}_m$ be the projection onto the first $m$ coordinates. We have

$$\|\mathbf{P}_{\mathcal{S}}\boldsymbol{z}^k\|_2^2 = \|\mathbf{U}^T\mathbf{P}_m\mathbf{U}\boldsymbol{z}^k\|_2^2 = \|\mathbf{P}_m\mathbf{U}\boldsymbol{z}^k\|_2, \tag{A.9}$$

Hence, we have

$$\frac{\|\mathbf{P}_{\mathcal{S}}\boldsymbol{z}^k\|_2^2}{\|\boldsymbol{z}^k\|_2^2} = \|\mathbf{P}_m\boldsymbol{y}\|_2^2, \tag{A.10}$$

where $\boldsymbol{y}$ is a random vector distributed uniformly in the unit sphere $\mathbb{S}^{d-1}$. We apply Lemma 2, and obtain

$$\mathbb{P}\left(\beta_1(m, \delta)\frac{m}{d} \leq \|\mathbf{P}_m\boldsymbol{y}\|_2^2 \leq \beta_2(m, \delta)\frac{m}{d}\right) \geq 1 - 2\delta. \tag{A.11}$$

Hence,

$$\mathbb{P}\left\{ \frac{1}{\beta_2(m, \delta)}\frac{d}{m} \leq \frac{\|\boldsymbol{r}_{\mathcal{S}}^k\|_2^2}{\|\boldsymbol{r}^k\|_2^2} \leq \frac{1}{\beta_1(m, \delta)}\frac{d}{m} \right\} \geq 1 - 2\delta. \tag{A.12}$$

Using the multi-class extension in Lemma 3, we conclude that

$$\mathbb{P}\left\{ \zeta_1(m, \delta)\frac{d}{m} \leq \frac{\|\boldsymbol{r}_{\mathcal{S}}^*\|_2^2}{\|\boldsymbol{r}^*\|_2^2} \leq \zeta_2(m, \delta)\frac{d}{m} \right\} \geq 1 - 2(L + 1)\delta. \tag{A.13}$$

$\square$

**Lemma 3** (Binary case to multiclass)**.** *Assume that, for all* $k \in \{1, \ldots, L]\} \backslash \{\hat{k}(\boldsymbol{x}_0)\}$

$$\mathbb{P}\left(l \leq \frac{\|\boldsymbol{r}_{\mathcal{S}}^k\|_2}{\|\boldsymbol{r}^k\|_2} \leq u\right) \geq 1 - \delta. \tag{A.14}$$

*Then, we have*

$$\mathbb{P}\left(l \leq \frac{\|\boldsymbol{r}_{\mathcal{S}}^*\|_2}{\|\boldsymbol{r}^*\|_2} \leq u\right) \geq 1 - (L+1)\delta. \tag{A.15}$$

*Proof.* Let $p := \arg\min_i \|\boldsymbol{r}^i\|_2$. Note that we have $\mathbb{P}\left(\frac{\|\boldsymbol{r}_{\mathcal{S}}^*\|_2}{\|\boldsymbol{r}^*\|_2} \geq u\right) \leq \mathbb{P}\left(\frac{\|\boldsymbol{r}_{\mathcal{S}}^p\|_2}{\|\boldsymbol{r}^p\|_2} \geq u\right) \leq \delta$. Moreover, we use a union bound to bound the the other bad event probability:

$$\mathbb{P}\left(\frac{\|\boldsymbol{r}_{\mathcal{S}}^*\|_2}{\|\boldsymbol{r}^*\|_2} \leq l\right) \leq \mathbb{P}\left(\bigcup_k \left\{\frac{\|\boldsymbol{r}_{\mathcal{S}}^k\|_2}{\|\boldsymbol{r}^k\|_2} \leq l\right\}\right) \leq L\delta, \tag{A.16}$$

$$\tag{A.17}$$

We conclude by using the fact that

$$\mathbb{P}\left(l \leq \frac{\|\boldsymbol{r}_{\mathcal{S}}^*\|_2}{\|\boldsymbol{r}^*\|_2} \leq u\right) = 1 - \mathbb{P}\left(\frac{\|\boldsymbol{r}_{\mathcal{S}}^*\|_2}{\|\boldsymbol{r}^*\|_2} \leq l\right) - \mathbb{P}\left(\frac{\|\boldsymbol{r}_{\mathcal{S}}^*\|_2}{\|\boldsymbol{r}^*\|_2} \geq u\right). \tag{A.18}$$

$\square$

## A.2 Proof of Theorem 2 and Corollary 1 (nonlinear classifiers)

First, we present an important geometric lemma and then use it to bound $\|\boldsymbol{r}_{\mathcal{S}}^*\|_2$. For the sake of the general readability of the section, some auxiliary results are given in Section A.3.

In the following result, we show that, when the curvature of a planar curve is constant and sufficiently small, the distance between a point $\boldsymbol{x}$ and the curve at a specific direction $\theta$ is well approximated by the distance between $\boldsymbol{x}$ and a straight line (see Fig. 1 for an illustration).

**Lemma 4.** *Let $\gamma$ be a planar curve of constant curvature $\kappa$. We denote by $r$ the distance between a point $\boldsymbol{x}$ and the curve $\gamma$. Denote moreover by $\mathcal{T}$ the tangent to $\gamma$ at the closest point to $\boldsymbol{x}$ (see Fig. 1). Let $\theta$ be the angle between $\boldsymbol{u}$ and $\boldsymbol{v}$ as depicted in Fig. 1. We assume that $r\kappa < 1$. We have*

$$-C_1 r\kappa \tan^2(\theta) \leq \frac{\|\boldsymbol{x}_\gamma - \boldsymbol{x}\|_2}{\|\boldsymbol{u}\|_2} - 1 \tag{A.19}$$

*Moreover, if*

$$\tan^2(\theta) \leq \frac{0.2}{r\kappa},$$

*then, the following upper bound holds*

$$\frac{\|\boldsymbol{x}_\gamma - \boldsymbol{x}\|_2}{\|\boldsymbol{u}\|_2} - 1 \leq C_2 r\kappa \tan^2(\theta). \tag{A.20}$$

*We can set $C_1 = 0.625$ and $C_2 = 2.25$.*

*Proof of upper bound.* We consider two distinct cases for the curve $\gamma$. In the case where $\gamma$ is concave-shaped (Fig. 1, right figure), we have

$$\frac{\|\boldsymbol{x}_\gamma - \boldsymbol{x}\|_2}{\|\boldsymbol{u}\|_2} \leq 1,$$

and the upper bound in Eq. (A.20) directly holds. We therefore focus on the case where $\gamma$ is convex-shaped as illustrated in the left figure of Fig. 1. Define $R := 1/\kappa$, one can write using simple geometric inspection

$$R^2 = \sin(\theta) r'^2 + (R + r - r' \cos(\theta))^2, \tag{A.21}$$

Figure 1: Bounding $\|\boldsymbol{x}_\gamma - \boldsymbol{x}\|_2$ in terms of $\kappa$.

where $r' = \|\boldsymbol{x}_\gamma - \boldsymbol{x}\|_2$. The discriminant of the second order equation (with variable $r'$) is equal to

$$\Delta = 4\left((R+r)^2 \cos^2(\theta) - (2rR + r^2)\right).$$

We have $\Delta \geq 0$ as $\theta$ satisfies the two assumptions $\tan^2(\theta) \leq 0.2R/r$ and $r/R < 1$. The smallest solution of this second order equation is given as follows

$$r' = (R+r)\cos(\theta) - \sqrt{(R+r)^2 \cos^2(\theta) - 2Rr - r^2}. \tag{A.22}$$

Using some simple algebraic manipulations, we obtain

$$r' = \frac{r}{\cos(\theta)}\left(\left(\frac{R}{r} + 1\right)\cos^2(\theta) - \frac{R}{r}\cos^2(\theta)\sqrt{1 - \tan^2(\theta)\frac{2Rr + r^2}{R^2}}\right). \tag{A.23}$$

Using the inequality in Lemma 7 together with the two assumptions, we get

$$r' \leq \frac{r}{\cos(\theta)}\left(\cos^2(\theta) + \frac{R}{r}\cos^2(\theta)\tan^2(\theta)\left(\frac{2Rr + r^2}{2R^2}\right)\right.$$
$$\left. + \frac{R}{r}\cos^2(\theta)\tan^4(\theta)\left(\frac{2Rr + r^2}{2R^2}\right)^2\right). \tag{A.24}$$

With simple trigonometric identities, the above expression can be simplified to

$$r' \leq \frac{r}{\cos(\theta)}\left(1 + \frac{r}{R}\left(\frac{\sin^2(\theta)}{2} + \frac{\sin^4(\theta)}{\cos^2(\theta)}\left(1 + \frac{r}{2R}\right)^2\right)\right). \tag{A.25}$$

We expand this quantity, and obtain

$$r' \leq \frac{r}{\cos(\theta)}\left(1 + \left(\frac{\sin^2(\theta)}{2} + \frac{\sin^4(\theta)}{\cos^2(\theta)}\right)\frac{r}{R} + \frac{\sin^4(\theta)}{\cos^2(\theta)}\frac{r^2}{R^2} + \frac{\sin^4(\theta)}{4\cos^2(\theta)}\frac{r^3}{R^3}\right). \tag{A.26}$$

Since $\sin^2(\theta)\tan^2(\theta) = \tan^2(\theta) - \sin^2(\theta)$, we have

$$r' \leq \frac{r}{\cos(\theta)}\left(1 + \tan^2(\theta)\left(\frac{r}{R} + \frac{r^2}{R^2} + \frac{r^3}{4R^3}\right)\right). \tag{A.27}$$

According to the assumptions $r/R < 1$, therefore

$$r' \leq \frac{r}{\cos(\theta)}\left(1 + 2.25\tan^2(\theta)\frac{r}{R}\right). \tag{A.28}$$

Since $r/\cos(\theta) = \|\boldsymbol{u}\|_2$, one can finally conclude on the upper bound

$$\frac{\|\boldsymbol{x}_\gamma - \boldsymbol{x}\|_2}{\|\boldsymbol{u}\|_2} - 1 \leq 2.25r\kappa\tan^2(\theta). \tag{A.29}$$

□

*Proof of lower bound.* When the curve is convex shaped (Fig. 1 left), we have $\|\boldsymbol{x}_\gamma - \boldsymbol{x}\|_2 \geq \|\boldsymbol{u}\|_2$, and the desired lower bound holds. We focus therefore on the case where $\gamma$ has a concave shape,

and coincides with with $\gamma_2$ (see Fig. 1 right). The following equation holds using simple geometric arguments

$$R^2 = \sin(\theta)r'^2 + (R - r + r'\cos(\theta))^2. \tag{A.30}$$

where $r' = \|\boldsymbol{x}_\gamma - \boldsymbol{x}\|_2$. Solving this second order equation gives

$$r' = -(R - r)\cos(\theta) + \sqrt{(R - r)^2\cos^2(\theta) - r^2 + 2Rr}. \tag{A.31}$$

After some algebraic manipulations, we get

$$r' = \frac{r}{\cos(\theta)}\left(-\left(\frac{R}{r} - 1\right)\cos^2(\theta) + \frac{R}{r}\cos^2(\theta)\sqrt{1 + \tan^2(\theta)\frac{2Rr - r^2}{R^2}}\right). \tag{A.32}$$

Using the inequality in Lemma 8, together with the fact that $r\kappa < 1$, we obtain

$$
\begin{aligned}
r' \geq \frac{r}{\cos(\theta)}\bigg( &\cos^2(\theta) + \frac{R}{r}\cos^2(\theta)\tan^2(\theta)\left(\frac{2Rr - r^2}{2R^2}\right) \\
&- \frac{R}{r}\frac{\cos^2(\theta)\tan^4(\theta)}{2}\left(\frac{2Rr - r^2}{2R^2}\right)^2\bigg).
\end{aligned}
\tag{A.33}
$$

Using simple trigonometric identities, the above expression is simplified to

$$r' \geq \frac{r}{\cos(\theta)}\left(1 + \frac{r}{R}\left(-\frac{\sin^2(\theta)}{2} - \frac{\sin^4(\theta)}{2\cos^2(\theta)}\left(1 - \frac{r}{2R}\right)^2\right)\right). \tag{A.34}$$

When expanding it, we obtain

$$r' \geq \frac{r}{\cos(\theta)}\left(1 - \left(\frac{\sin^2(\theta)}{2} + \frac{\sin^4(\theta)}{2\cos^2(\theta)}\right)\frac{r}{R} + \frac{\sin^4(\theta)}{2\cos^2(\theta)}\frac{r^2}{R^2} - \frac{\sin^4(\theta)}{8\cos^2(\theta)}\frac{r^3}{R^3}\right). \tag{A.35}$$

Since $\sin^2(\theta)\tan^2(\theta) = \tan^2(\theta) - \sin^2(\theta)$, we have

$$r' \geq \frac{r}{\cos(\theta)}\left(1 - \tan^2(\theta)\left(\frac{r}{2R} + \frac{r^3}{8R^3}\right)\right). \tag{A.36}$$

Using again the assumption $r/R < 1$, we obtain

$$r' \geq \frac{r}{\cos(\theta)}\left(1 - 0.625\tan^2(\theta)\frac{r}{R}\right). \tag{A.37}$$

Since $r/\cos(\theta) = \|\boldsymbol{u}\|_2$, one can rewrite it as

$$\frac{\|\boldsymbol{x}_\gamma - \boldsymbol{x}\|_2}{\|\boldsymbol{u}\|_2} - 1 \geq -0.625 r\kappa\tan^2(\theta), \tag{A.38}$$

which completes the proof. $\qquad\square$

We now use the previous lemma to bound the semi-random robustness of the classifier, i.e. $\|\boldsymbol{r}_{\mathcal{S}}^k\|_2$, to the worst-case robustness $\|\boldsymbol{r}^k\|_2$ in the case where the curvature is sufficiently small.

**Theorem 2.** *Let $\mathcal{S}$ be a random $m$-dimensional subspace of $\mathbb{R}^d$. Define $\alpha := \sqrt{m/d}$, and let $\kappa := \kappa(\mathscr{B}_k)$. Assuming that $\kappa \leq \frac{C\alpha^2}{\zeta_2(m,\delta)\|\boldsymbol{r}^k\|_2}$, the following inequalities hold between $\|\boldsymbol{r}_{\mathcal{S}}^k\|_2$ and the worst-case perturbation $\|\boldsymbol{r}^k\|_2$*

$$\frac{\zeta_1(m,\delta)}{\alpha^2}\left(1 - \frac{C_1\|\boldsymbol{r}^k\|_2\kappa\zeta_2(m,\delta)}{\alpha^2}\right)^2 \leq \frac{\|\boldsymbol{r}_{\mathcal{S}}^k\|_2^2}{\|\boldsymbol{r}^k\|_2^2} \leq \frac{\zeta_2(m,\delta)}{\alpha^2}\left(1 + \frac{C_2\|\boldsymbol{r}^k\|_2\kappa\zeta_2(m,\delta)}{\alpha^2}\right)^2$$
$$\tag{A.39}$$

*with probability larger than $1 - 4\delta$. The constants can be taken $C = 0.2, C_1 = 0.625, C_2 = 2.25$.*

Figure 2: Left: To prove the upper bound, we consider a ball $\mathcal{B}$ included in $\mathcal{R}_k$ that intersects with the boundary at $\boldsymbol{x}^*$. Upper bounds on $\|r_{\mathcal{S}}^k\|_2$ derived when the boundary is $\partial\mathcal{B}$ are also valid upper bounds for the real boundary $\mathscr{B}_k$. Right: Normal section to the decision boundary $\mathscr{B}_k = \partial\mathcal{B}$ along the normal plane $\mathcal{U} = \text{span}\left(r_{\mathcal{S}}^{\mathcal{T}}, r^k\right)$. We denote by $\gamma$ the normal section of boundary $\mathscr{B}_k$, along the plane $\mathcal{U}$, and by $\mathcal{T}_{\boldsymbol{x}^*}\mathscr{B}_k$ the tangent space to the sphere $\partial\mathcal{B}$ at $\boldsymbol{x}^*$.

*Proof of upper bound.* Denote by $\boldsymbol{x}^*$ the point belonging to the boundary $\mathscr{B}_k$ that is closest to the original data point $\boldsymbol{x}_0$. By definition of the curvature $\kappa$, there exists a point $\boldsymbol{z}^*$ such that the ball $\mathcal{B}$ centered at $\boldsymbol{z}^*$ and of radius $1/\kappa = \|\boldsymbol{z}^* - \boldsymbol{x}^*\|_2$ is inscribed in the region $\mathcal{R}_k = \{x \in \mathbb{R}^d : f_k(\boldsymbol{x}) > f_{\hat{k}(\boldsymbol{x}_0)}(\boldsymbol{x})\}$ (see Fig. 2 (a)).[2]

Observe that the worst-case perturbation along any subspace $\mathcal{S}$ that reaches the ball $\mathcal{B}$ is larger than the perturbation along $\mathcal{S}$ that reaches the region $\mathcal{R}_k$, as $\mathcal{B} \subseteq \mathcal{R}_k$. Therefore, any upper bound derived when the boundary is the sphere of radius $1/\kappa$; i.e., $\mathscr{B}_k = \partial\mathcal{B}$ is also a valid upper bound for boundary $\mathscr{B}_k$ (see Fig. 2 (a)). It is therefore sufficient to derive an upper bound in the worst case scenario where the boundary $\mathscr{B}_k = \partial\mathcal{B}$, and we consider this case for the remainder of the proof of the upper bound.

We now consider the linear classifier whose boundary is tangent to $\mathscr{B}_k$ at $\boldsymbol{x}^*$. For the random subspace $\mathcal{S}$, we denote by $r_{\mathcal{S}}^{\mathcal{T}}$ the worst-case subspace perturbation for this linear classifier. We then focus on the intersection between the boundary $\mathscr{B}_k$ and the two-dimensional plane $\mathcal{U}$ spanned by the vectors $r^k$ and $r_{\mathcal{S}}^{\mathcal{T}}$. This *normal* section of the boundary cuts the ball $\mathcal{B}$ through its center as the tangent spaces of the decision boundary and the ball coincide. See Fig. 2 for a clarifying figure of this two-dimensional cross-section. We define the angle $\hat{\theta}$ as denoted in Fig. 2, such that $\cos(\hat{\theta}) = \frac{\|r^k\|_2}{\|r_{\mathcal{S}}^{\mathcal{T}}\|_2}$.

We apply our result on linear classifiers in Theorem 1 for the tangent classifier. We have

$$\frac{1}{\cos(\hat{\theta})^2} = \frac{\|r_{\mathcal{S}}^{\mathcal{T}}\|_2^2}{\|r^k\|_2^2} \leq \frac{1}{\alpha^2}\zeta_2(m, \delta), \tag{A.40}$$

with probability exceeding $1 - 2\delta$. Hence, using $\tan^2(\hat{\theta}) \leq (\cos^2(\hat{\theta}))^{-1}$ and the assumption of the theorem, we deduce that

$$\tan^2(\hat{\theta}) \leq \frac{1}{\alpha^2}\zeta_2(m, \delta) \leq \frac{0.2}{\kappa\|r^k\|_2},$$

with probability exceeding $1 - 2\delta$. Note moreover that

$$\|r^k\|_2\kappa \leq \frac{0.2\alpha^2}{\zeta_2(m, \delta)} < 1.$$

Hence, the assumptions of Lemma 4 hold with probability larger than $1 - 2\delta$. Using the notations of Fig. 2, we therefore obtain from Lemma 4

$$\frac{\|\boldsymbol{x}_\gamma - \boldsymbol{x}_0\|_2}{\|r_{\mathcal{S}}^{\mathcal{T}}\|_2} - 1 \leq C_2\kappa\|r^k\|_2\tan^2(\hat{\theta}) \tag{A.41}$$

with probability larger than $1 - 2\delta$.

Observe that $\|\boldsymbol{x}_\gamma - \boldsymbol{x}_0\|_2 \geq \|\boldsymbol{r}_\mathcal{S}^k\|_2$, and that $\tan^2(\hat{\theta}) \leq \frac{\|\boldsymbol{r}_\mathcal{S}^\mathcal{T}\|_2^2}{\|\boldsymbol{r}^k\|_2^2}$. Hence, we obtain by re-writing Eq. (A.41)

$$\mathbb{P}\left(\frac{\|\boldsymbol{r}_\mathcal{S}^k\|_2^2}{\|\boldsymbol{r}^k\|_2^2} \leq \left\{1 + C_2\kappa\|\boldsymbol{r}^k\|_2\frac{\|\boldsymbol{r}_\mathcal{S}^\mathcal{T}\|_2^2}{\|\boldsymbol{r}^k\|_2^2}\right\}^2 \frac{\|\boldsymbol{r}_\mathcal{S}^\mathcal{T}\|_2^2}{\|\boldsymbol{r}^k\|_2^2}\right) \geq 1 - 2\delta. \tag{A.42}$$

Using the inequality in Eq. (A.40), we obtain

$$\mathbb{P}\left(\frac{\|\boldsymbol{r}_\mathcal{S}^k\|_2^2}{\|\boldsymbol{r}^k\|_2^2} \leq \left\{1 + C_2\kappa\|\boldsymbol{r}^k\|_2\frac{\zeta_2(m,\delta)}{\alpha^2}\right\}^2 \frac{\zeta_2(m,\delta)}{\alpha^2}\right) \geq 1 - 2\delta,$$

which concludes the proof of the upper bound. $\qquad\square$

*Proof of the lower bound.* We now consider the ball $\mathcal{B}'$ of center $\boldsymbol{z}^*$ and radius $1/\kappa = \|\boldsymbol{z}^* - \boldsymbol{x}^*\|_2$ that is included in the region $\mathcal{R}_{\hat{k}(\boldsymbol{x}_0)}$. Since the ball $\mathcal{B}'$ is, by definition, included in the region $\mathcal{R}_{\hat{k}(\boldsymbol{x}_0)}$, the worst-case scenario for the lower bound on $\|\boldsymbol{r}_\mathcal{S}^k\|_2$ occurs whenever the decision boundary $\mathscr{B}_k$ coincides with the ball $\mathcal{B}'$ (see Fig. 3 (a)). We consider this case in the remainder of the proof.

To derive the lower bound, we consider the cross-section $\mathcal{U}'$ spanned by the vectors $\boldsymbol{r}_\mathcal{S}^k$ and $\boldsymbol{r}^k$ (Fig. 3 (b)). We have $\|\boldsymbol{r}^k\|_2\kappa < 1$; using the lower bound of Lemma 4, we obtain

$$-C_1\kappa\|\boldsymbol{r}^k\|_2\tan^2(\tilde{\theta}) \leq \frac{\|\boldsymbol{r}_\mathcal{S}^k\|_2}{\|\boldsymbol{x}_\mathcal{T} - \boldsymbol{x}_0\|_2} - 1 \tag{A.43}$$

for any $\mathcal{S}$. Observe moreover that

$$\tan^2(\tilde{\theta}) \leq \frac{1}{\cos(\tilde{\theta})^2} = \frac{\|\boldsymbol{x}_\mathcal{T} - \boldsymbol{x}_0\|_2^2}{\|\boldsymbol{r}^k\|_2^2}.$$

Hence, the following bound holds:

$$\frac{\|\boldsymbol{x}_\mathcal{T} - \boldsymbol{x}_0\|_2^2}{\|\boldsymbol{r}^k\|_2^2}\left(1 - C_1\kappa\|\boldsymbol{r}^k\|_2\frac{\|\boldsymbol{x}_\mathcal{T} - \boldsymbol{x}_0\|_2^2}{\|\boldsymbol{r}^k\|_2^2}\right)^2 \leq \frac{\|\boldsymbol{r}_\mathcal{S}^k\|_2^2}{\|\boldsymbol{r}^k\|_2^2}.$$

Let $\boldsymbol{r}_\mathcal{S}^\mathcal{T}$ denote the worst-case perturbation belonging to subspace $\mathcal{S}$ for the *linear* classifier $\mathcal{T}_{x^*}\mathscr{B}_k$. It is not hard to see that $\boldsymbol{r}_\mathcal{S}^\mathcal{T}$ is *collinear* to $\boldsymbol{r}_\mathcal{S}^k$ (see Lemma 6 for a proof). Hence, we have $\boldsymbol{r}_\mathcal{S}^\mathcal{T} = \boldsymbol{x}_\mathcal{T} - \boldsymbol{x}_0$. By applying our result on linear classifiers in Theorem 1 for the tangent classifier $\mathcal{T}_{x^*}\mathscr{B}_k$, we have:

$$\mathbb{P}\left(\frac{\zeta_1(m,\delta)}{\alpha^2} \leq \frac{\|\boldsymbol{r}_\mathcal{S}^\mathcal{T}\|_2^2}{\|\boldsymbol{r}^k\|_2^2} \leq \frac{\zeta_2(m,\delta)}{\alpha^2}\right) \geq 1 - 2\delta.$$

We therefore conclude that

$$\mathbb{P}\left(\frac{\zeta_1(m,\delta)}{\alpha^2}\left\{1 - C_1\kappa\|\boldsymbol{r}^k\|_2\frac{\zeta_2(m,\delta)}{\alpha^2}\right\}^2 \leq \frac{\|\boldsymbol{r}_\mathcal{S}^k\|_2^2}{\|\boldsymbol{r}^k\|_2^2}\right) \geq 1 - 2\delta,$$

which concludes the proof of the lower bound.

$\qquad\square$

The goal is now to extend the previous result, derived for binary classifiers, to the multiclass classification case. To do so, we show the following lemma.

**Lemma 5** (Binary case to multiclass). *Let $p = \arg\min_i \|\boldsymbol{r}^i\|_2$. Define the deterministic set*

$$A = \left\{k : \|\boldsymbol{r}^k\|_2 \geq 1.45\sqrt{\zeta_2(m,\delta)}\sqrt{\frac{d}{m}}\|\boldsymbol{r}^*\|_2\right\}. \tag{A.44}$$

*Assume that, for all $k \in A^c$, we have*

$$\mathbb{P}\left(l \leq \frac{\|\boldsymbol{r}_\mathcal{S}^k\|_2}{\|\boldsymbol{r}^k\|_2} \leq u\right) \geq 1 - \delta. \tag{A.45}$$

Figure 3: Left: To prove the lower bound, we consider a ball $\mathcal{B}'$ included in $\mathcal{R}_{\hat{k}(\boldsymbol{x}_0)}$ that intersects with the boundary at $\boldsymbol{x}^*$. Lower bounds on $\|\boldsymbol{r}_{\mathcal{S}}^k\|_2$ derived when the boundary is the sphere $\partial\mathcal{B}'$ are also valid lower bounds for the real boundary $\mathscr{B}_k$. Right: Cross section of the problem along the plane $\mathcal{U}' = \text{span}\left(\boldsymbol{r}_{\mathcal{S}}^k, \boldsymbol{r}^k\right)$. $\gamma$ denotes the normal section of $\mathscr{B}_k = \mathcal{B}'$ along the plane $\mathcal{U}'$.

*and that*

$$\mathbb{P}\left(\|\boldsymbol{r}_{\mathcal{S}}^p\|_2 \geq 1.45\sqrt{\zeta_2(m,\delta)}\sqrt{\frac{d}{m}}\|\boldsymbol{r}^*\|_2\right) \leq t. \tag{A.46}$$

*Then, we have*

$$\mathbb{P}\left(l \leq \frac{\|\boldsymbol{r}_{\mathcal{S}}^*\|_2}{\|\boldsymbol{r}^*\|_2} \leq u\right) \geq 1 - (L+1)\delta - t. \tag{A.47}$$

*Proof.* Note first that

$$\mathbb{P}\left(\frac{\|\boldsymbol{r}_{\mathcal{S}}^*\|_2}{\|\boldsymbol{r}^*\|_2} \geq u\right) \leq \mathbb{P}\left(\left\{\frac{\|\boldsymbol{r}_{\mathcal{S}}^p\|_2}{\|\boldsymbol{r}^p\|_2} \geq u\right\}\right) \leq \delta. \tag{A.48}$$

We now focus on bounding the other bad event probability $\mathbb{P}\left(\frac{\|\boldsymbol{r}_{\mathcal{S}}^*\|_2}{\|\boldsymbol{r}^*\|_2} \leq l\right)$. We have

$$\mathbb{P}\left(\frac{\|\boldsymbol{r}_{\mathcal{S}}^*\|_2}{\|\boldsymbol{r}^*\|_2} \leq l\right) = \mathbb{P}\left(\min_{k \notin A}\|\boldsymbol{r}_{\mathcal{S}}^k\|_2 = \|\boldsymbol{r}_{\mathcal{S}}^*\|_2, \frac{\|\boldsymbol{r}_{\mathcal{S}}^*\|_2}{\|\boldsymbol{r}^*\|_2} \leq l\right) + \mathbb{P}\left(\min_{k \in A}\|\boldsymbol{r}_{\mathcal{S}}^k\|_2 = \|\boldsymbol{r}_{\mathcal{S}}^*\|_2, \frac{\|\boldsymbol{r}_{\mathcal{S}}^*\|_2}{\|\boldsymbol{r}^*\|_2} \leq l\right) \tag{A.49}$$

The first probability can be bounded as follows:

$$\mathbb{P}\left(\min_{k \notin A}\|\boldsymbol{r}_{\mathcal{S}}^k\|_2 = \|\boldsymbol{r}_{\mathcal{S}}^*\|_2, \frac{\|\boldsymbol{r}_{\mathcal{S}}^*\|_2}{\|\boldsymbol{r}^*\|_2} \leq l\right) \leq \mathbb{P}\left(\bigcup_{k \notin A}\frac{\|\boldsymbol{r}_{\mathcal{S}}^*\|_2}{\|\boldsymbol{r}^*\|_2} \leq l\right) \leq L\delta. \tag{A.50}$$

The second probability can also be bounded in the following way

$$\mathbb{P}\left(\min_{k \in A}\|\boldsymbol{r}_{\mathcal{S}}^k\|_2 = \|\boldsymbol{r}_{\mathcal{S}}^*\|_2, \frac{\|\boldsymbol{r}_{\mathcal{S}}^*\|_2}{\|\boldsymbol{r}^*\|_2} \leq l\right) \leq \mathbb{P}\left(\min_{k \in A}\|\boldsymbol{r}_{\mathcal{S}}^k\|_2 = \|\boldsymbol{r}_{\mathcal{S}}^*\|_2\right) = \mathbb{P}\left(\exists k \in A, \|\boldsymbol{r}_{\mathcal{S}}^k\|_2 \leq \|\boldsymbol{r}_{\mathcal{S}}^*\|_2\right). \tag{A.51}$$

Observe that, for $k \in A$, we have $\|\boldsymbol{r}_{\mathcal{S}}^k\|_2 \geq \|\boldsymbol{r}^k\|_2 \geq 1.45\sqrt{\zeta_2(m,\delta)}\sqrt{\frac{d}{m}}\|\boldsymbol{r}^*\|_2$. Hence, we conclude that

$$\mathbb{P}\left(\min_{k \in A}\|\boldsymbol{r}_{\mathcal{S}}^k\|_2 = \|\boldsymbol{r}_{\mathcal{S}}^*\|_2, \frac{\|\boldsymbol{r}_{\mathcal{S}}^*\|_2}{\|\boldsymbol{r}^*\|_2} \leq l\right) \leq \mathbb{P}\left(1.45\sqrt{\zeta_2(m,\delta)}\sqrt{\frac{d}{m}}\|\boldsymbol{r}^*\|_2 \leq \|\boldsymbol{r}_{\mathcal{S}}^*\|_2\right) \tag{A.52}$$

$$\leq \mathbb{P}\left(1.45\sqrt{\zeta_2(m,\delta)}\sqrt{\frac{d}{m}}\|\boldsymbol{r}^*\|_2 \leq \|\boldsymbol{r}_{\mathcal{S}}^p\|_2\right) \leq t. \tag{A.53}$$

$\square$

**Corollary 1.** *Let $\mathcal{S}$ be a random $m$-dimensional subspace of $\mathbb{R}^d$. Assume that, for all $k \notin A$, we have*

$$\kappa(\mathscr{B}_k)\|\boldsymbol{r}^k\|_2 \leq \frac{0.2}{\zeta_2(m,\delta)}\frac{m}{d} \tag{A.54}$$

*Then, we have*

$$0.875\sqrt{\zeta_1(m,\delta)}\sqrt{\frac{d}{m}} \leq \frac{\|\boldsymbol{r}^*_{\mathcal{S}}\|_2}{\|\boldsymbol{r}^*\|_2} \leq 1.45\sqrt{\zeta_2(m,\delta)}\sqrt{\frac{d}{m}} \tag{A.55}$$

*with probability larger than $1 - 4(L+2)\delta$.*

*Proof.* Using Theorem 2, we have that for all $k \notin A$, the result in Eq. (A.39) holds. We simplify the result with the assumption $\kappa(\mathscr{B}_k)\|\boldsymbol{r}\|_2 \leq \frac{0.2}{\zeta_2(m,\delta)}\frac{m}{d}$. Hence, the bounds of Theorem 2 are given as follows

$$\frac{\zeta_1(m,\delta)}{\alpha^2}\left(1 - 0.2C_1\right)^2 \leq \frac{\|\boldsymbol{r}^k_{\mathcal{S}}\|_2^2}{\|\boldsymbol{r}^k\|_2^2} \leq \frac{\zeta_2(m,\delta)}{\alpha^2}\left(1 + 0.2C_2\right)^2, \tag{A.56}$$

which leads to the following bounds:

$$\zeta_1(m,\delta)\frac{d}{m}0.875^2 \leq \frac{\|\boldsymbol{r}^k_{\mathcal{S}}\|_2^2}{\|\boldsymbol{r}^k\|_2^2} \leq \zeta_2(m,\delta)\frac{d}{m}1.45^2, \tag{A.57}$$

with probability exceeding $1 - 4\delta$.

By using Lemma 5, together with the fact that $t = \delta$, we obtain

$$\mathbb{P}\left(0.875\sqrt{\zeta_1(m,\delta)}\sqrt{\frac{d}{m}} \leq \frac{\|\boldsymbol{r}^*_{\mathcal{S}}\|_2}{\|\boldsymbol{r}^*\|_2} \leq 1.45\sqrt{\zeta_2(m,\delta)}\sqrt{\frac{d}{m}}\right) \geq 1 - 4(L+2)\delta, \tag{A.58}$$

which concludes the proof. $\qquad\square$

### A.3 Useful results

Figure 4: The worst-case perturbation in the subspace $\mathcal{S}$ when the decision boundary is $\partial\mathcal{B}$ and $\mathcal{T}_{\boldsymbol{x}^*}(\partial\mathcal{B})$ (denoted respectively by $\boldsymbol{r}^{\mathcal{B}}_{\mathcal{S}}$ and $\boldsymbol{r}^{\mathcal{T}}_{\mathcal{S}}$) are collinear.

**Lemma 6.** *Let $\boldsymbol{x}_0 \in \mathbb{R}^d$, and $\boldsymbol{x}^*$ denote the closest point to $\boldsymbol{x}_0$ on the sphere $\partial\mathcal{B}$ (see Fig. 4). Let $\mathcal{T}_{\boldsymbol{x}^*}(\partial\mathcal{B})$ be the tangent space to $\partial\mathcal{B}$ at $\boldsymbol{x}^*$. For an arbitrary subspace $\mathcal{S}$, let $\boldsymbol{r}^{\mathcal{T}}_{\mathcal{S}}$ and $\boldsymbol{r}^{\mathcal{B}}_{\mathcal{S}}$ denote the worst-case perturbations of $\boldsymbol{x}_0$ on the subspace $\mathcal{S}$, when the decision boundaries are respectively $\mathcal{T}_{\boldsymbol{x}^*}(\partial\mathcal{B})$ and $\partial\mathcal{B}$. Then, the two perturbations $\boldsymbol{r}^{\mathcal{T}}_{\mathcal{S}}$ and $\boldsymbol{r}^{\mathcal{B}}_{\mathcal{S}}$ are collinear.*

*Proof.* Assuming the center of the ball $\mathcal{B}$ is the origin, the points on the sphere $\partial\mathcal{B}$ satisfy equation: $\|\boldsymbol{x}\|_2 = R$, where $R$ denotes the radius. Hence, the perturbation $\boldsymbol{r}^{\mathcal{B}}_{\mathcal{S}}$ is given by

$$\boldsymbol{r}^{\mathcal{B}}_{\mathcal{S}} = \underset{\boldsymbol{r}\in\mathbb{R}^d}{\operatorname{argmin}}\|\boldsymbol{r}\|_2^2 \text{ such that } \|\boldsymbol{x}_0 + \mathbf{P}_{\mathcal{S}}\boldsymbol{r}\|_2^2 = R^2. \tag{A.59}$$

By equating the gradient of Lagrangian of the above constrained optimization problem to zero, we obtain the following necessary optimality condition

$$\boldsymbol{r} + \lambda\mathbf{P}_{\mathcal{S}}(\boldsymbol{x}_0 + \mathbf{P}_{\mathcal{S}}\boldsymbol{r}) = 0.$$

It should further be noted that $\mathbf{P}_{\mathcal{S}} r_{\mathcal{S}}^{\mathcal{B}} = r_{\mathcal{S}}^{\mathcal{B}}$. Indeed, if $r_{\mathcal{S}}^{\mathcal{B}}$ had a component orthogonal to $\mathcal{S}$, the projection of $r_{\mathcal{S}}^{\mathcal{B}}$ onto $\mathcal{S}$ would have strictly lower $\ell_2$ norm, while still satisfying the condition in Eq.(A.59). Hence, the necessary condition of optimality becomes

$$(1+\lambda)\boldsymbol{r} + \lambda \mathbf{P}_{\mathcal{S}} \boldsymbol{x}_0 = 0,$$

from which we conclude that $r_{\mathcal{S}}^{\mathcal{B}}$ is collinear to $\mathbf{P}_{\mathcal{S}} \boldsymbol{x}_0$.

It should further be noted that $r_{\mathcal{S}}^{\mathcal{T}}$ can be computed in closed form, and is collinear to $\mathbf{P}_{\mathcal{S}}(\boldsymbol{x}^* - \boldsymbol{x}_0)$, which is itself collinear to $\boldsymbol{x}_0$, as the the center of the ball was assumed to be the origin. This concludes the proof. $\qquad\square$

**Lemma 7.** *If $x \in [0, 2(\sqrt{2}-1)]$,*

$$\sqrt{1-x} \geq 1 - \frac{x}{2} - \frac{x^2}{4}. \tag{A.60}$$

**Lemma 8.** *If $x \geq 0$,*

$$\sqrt{1+x} \geq 1 + \frac{x}{2} - \frac{x^2}{8}. \tag{A.61}$$

## Footnotes

[2]For a fixed point $\boldsymbol{x}^*$ on the boundary, the maximal radius $1/\kappa$ might not be achieved. To prove the result in the general case where the supremum is not achieved, one can consider instead a sequence $(\kappa_n)_n$ converging to $\kappa$, such that the balls of radius $1/\kappa_n$ and intersecting the boundary at $\boldsymbol{x}^*$ are included in $\mathcal{R}_k$. The same proof and results follow by taking the limit on the bounds derived with ball of radius $1/\kappa_n$.

# References

[1] Dasgupta, S. and Gupta, A. (2003). An elementary proof of a theorem of johnson and lindenstrauss. *Random Structures & Algorithms*, 22(1):60–65.