[Reviews · NeurIPS 2016]

Reviewer 1

Summary

This paper offers a thorough analysis of the effect of both worse-case (adversarial) and random noise in machine learning classifiers. It derives bounds that precisely describe the robustness of classifiers in function of the curvature of the decision boundary. This leads to some surprisingly (at least to me) general conclusions: * For random noise, the robustness of classifiers behaves as sqrt(d) times the distance from the datapoint to the classification boundary (where d denotes the dimension of the data) provided the curvature of the decision boundary is sufficiently small. This corroborates the intuition that random noise is less of an issue for high-dimensional data. On the other hand, how do we know the curvature of decision boundaries for general classifiers? * For semi-random noise, defined as the worst possible perturbation possible a random subspace of dimension m, the robustness behaves as sqrt(d/m) times the distance to the boundary. Hence, it is quite possible to find small perturbations that cause data misclassification, even within strong constraints. These theoretical estimates are evaluated empirically on image recognition tasks using deep neural nets. Vice versa, it leads to some insight on the curvature of the decision boundary in deep neural nets. Overall, this is very solid work that will likely appeal to the NIPS community. If I have one comments, it is that this work should be extended to many other types of algorithms to demonstrate its claimed generality.

Qualitative Assessment

This is very solid work and the paper is overall very pleasant to read. My one comment would be that this work seems to open up a lot of interesting future work when applied to other learners besides deep neural nets (and in some cases, we know something about the curvature of the decision boundaries (e.g. decision trees) that would make for interesting analysis. Are the authors interested in pursuing this? Minor point: There is no reference in the paper to the proofs (in the supplementary materials). Make sure that they are easy to find in the final version.

Confidence in this Review

2-Confident (read it all; understood it all reasonably well)


Reviewer 2

Summary

The authors gave an evaluation of the robustness of classifiers via a semi-random noise regime which includes the random and worse-case noise condition. They gave some comments to related studies in robustness of especially non-linear classifiers and demonstrated where this work differs from those works. Through comparative experiments in estimating the performance of some state-of-the-art deep neural networks and datasets, they proposed that their bounds corresponds to the decision boundary, and showed that the classifiers are robust to random noise versus semi-random noise in high dimensional classification issues, if the boundary has a small curvature. Improving the robustness of learning neural networks like the CNN, may benefit from the proposed way of analysis.

Qualitative Assessment

Please see some feedbacks below: 1. The description of experimental datasets seems not be clear enough for the results in Table 1. 2. The caption of Figure 5 makes some confusion.

Confidence in this Review

1-Less confident (might not have understood significant parts)


Reviewer 3

Summary

This paper provides an interesting analysis of the robustness of classifiers to random versus worst-case noise perturbations. The authors also describe a semi-random noise perturbations where a worst case is found in a subspace of the input. The authors find that in the random case and when the curvature of the decision boundary is small the robustness is sqrt(d) times the distance of the datapoint to the classification boundary. In the semi-random case however, the robustness is sqrt(d/m) which shows that even when m is small compared to d, small perturbations can result in misclassification. The authors then empirically verify the theoretical estimates on actual networks and datasets.

Qualitative Assessment

How would the geometric constraints on the curvature be imposed during the training of the CNN to make it more robust to semi-random noise? Is there a way to incorporate this into the cost during training, or would the approach involve creating modified training examples as in Fig. 3c?

Confidence in this Review

1-Less confident (might not have understood significant parts)


Reviewer 4

Summary

The paper presents a quantitative analysis of robustness in linear and nonlinear classifiers to distortion. It tries to give precise boundaries in a general noise regime and prove its solidity by conducting some experiments for CNN classifiers. Authors claim that this study could "impose some geometric constraint ... during the training process" to improve the robustness of CNN to noise.

Qualitative Assessment

1. My first concern is about problem setting on L74 which introduce "equal" to the formulation. If authors consider fk(x) as unary potentials assigned to class k, "equal" means same unary for noisy and estimated classes. According to maximization in L70, "equality" would not make clear label estimation. 2. In noise regimes at L83-95, I could not realize the reasoning behind "uniform" random sampling. I'm just wondering Why this problem set-up needs uniformity in random sub-spaces, because it is not used anywhere inside the paper or supplementary materials. 3. There is a sub-section on affine classifiers and a reference on L137 to experimental section, but I could not see any affine model there. Actually, deep learning models are not generally affine in any sub-spaces and the results do not support the claimed affine boundaries. 4. In Theorem 2 at L171, C1 should also set zero to recover results for affine classifiers. 5. It seems to me that conducting experiments on noise, perturbation and fooling hidden message may be good evaluation for the theoretical formulation, but the main problem in deep learning for vision is light/pose/depth/color/compression variations in test samples.

Confidence in this Review

2-Confident (read it all; understood it all reasonably well)


Reviewer 5

Summary

a) This paper shows connections between the random/semi-random noise robustness and the worst-case noise robustness of classifiers (noted as ||r*||_2), given that the curvature of decision boundary is sufficiently small. For example, the random noise robustness is sqrt(d)||r*||_2, where d is the dimensionality of data. b) This paper provides empirical evidence that verify the provided connection.

Qualitative Assessment

The novelty of giving theoretical relations between random and semi-random noise to worst-case robustness for nonlinear classifiers does hold. However, this contribution is somewhat incremental. First, the authors assume that the curvature of decision boundary is well defined and sufficiently small. This assumption does not hold if the decision boundary is discontinuous. Second, this paper did not suggest a way to design classifiers that are robust to random/semi-random/worst-case noise.

Confidence in this Review

2-Confident (read it all; understood it all reasonably well)